

# Quantification of Magnetosphere - Ionosphere coupling timescales using mutual information: response of terrestrial radio emissions and ionospheric/magnetospheric currents

Alexandra Ruth Fogg[1], Caitriona M Jackman[1], Sandra C Chapman[2,3,4], James E Waters[5], Aisling Bergin[2], Laurent Lamy[6,7], Karine Issautier[6], Baptiste Cecconi[6], and Xavier Bonnin[6]

[1]School of Cosmic Physics, DIAS Dunsink Observatory, Dublin Institute for Advanced Studies, Dublin 15, Ireland
[2]CFSA, Physics Department, University of Warwick, UK
[3]Department of Mathematics and Statistics, University of Tromso, Norway
[4]ISSI, Bern, Switzerland
[5]Aix Marseille Univ, CNRS, CNES, LAM, Marseille, France
[6]LESIA, Observatoire de Paris, Université PSL, CNRS, Sorbonne Université, Université de Paris, France
[7]LAM, Pythéas, Aix Marseille Université, CNRS, CNES, 38 Rue Frédéric Joliot Curie, 13013 Marseille, France

**Correspondence:** Alexandra Ruth Fogg (arfogg@cp.dias.ie)

**Abstract.** Auroral kilometric radiation (AKR) is a terrestrial radio emission, excited by the same accelerated electrons which excite auroral emissions. Although it is well correlated with auroral and geomagnetic activity, the coupling timescales between AKR and different magnetospheric/ionospheric regions are yet to be determined. Estimation of these coupling timescales is non-trivial as a result of complex, non-linear processes which rarely occur in isolation. In this study, the mutual information between AKR intensity and different geomagnetic indices is used to assess the correlation between variables. Indices are shifted to different temporal lags relative to AKR intensity, and the lag at which the variables have the greatest shared information is found. This lag is interpreted as the coupling timescale. The AKR source region receives the effects of a shared driver before the auroral ionosphere. Conversely, the polar ionosphere reacts to a shared driver before the AKR source region. Bow shock IMF $B_Z$ is excited about an hour before AKR enhancements.

## 1 Introduction

Auroral kilometric radiation (AKR) is the strongest natural terrestrial radio emission, and is excited by the same electron precipitation which stimulates auroral emission in the ionosphere. AKR was first observed in the 1960s by Dunckel et al. (1970), and has been studied in detail since by instruments on board spacecraft including IMP 6 and 8, Hawkeye, Wind, GEOTAIL, POLAR, IMAGE, the Cluster array and Cassini (e.g., Green et al., 1977; Voots et al., 1977; Gurnett, 1974; Gallagher and D'Angelo, 1981; Desch et al., 1996; Kasaba et al., 1997; Hashimoto et al., 1998; Kurth et al., 1998; Green et al., 2003; Mutel et al., 2008; Lamy et al., 2010; Waters et al., 2021b; Fogg et al., 2022; Waters et al., 2022). With its main band appearing between 100 and 400 kHz, AKR is more broadly observed between 30 and 800 kHz (Gurnett, 1974; Benson and Calvert, 1979; Green and Gurnett, 1979; Benson et al., 1980; Huff et al., 1988), with powers up to $10^9$ W (e.g., Gurnett, 1974; Zhao et al., 2019). AKR observations have been shown to be well correlated with substorm activity (e.g., Morioka et al., 2011), and indeed





the Auroral Electrojet (AE) index (Voots et al., 1977; Gurnett, 1974; Dunckel et al., 1970). Furthermore, Waters et al. (2022) showed enhancements in AKR power and frequency expansion at substorm onset over a decade of Wind/WAVES data. Finally, AKR is an excellent indicator of geomagnetic disturbance, particularly at nightside local times (LTs) (Zhao et al., 2019).

AKR is generated in a region of low plasma density, known as the auroral plasma cavity (Calvert, 1981; Ergun et al., 1998; Hilgers, 1992; Johnson et al., 2001), within which precipitating electrons are observed (Green and Gurnett, 1979; Ergun

et al., 1998). Energetic electrons within these auroral acceleration regions are propelled down magnetic field lines towards the ionosphere, often following a dipolarisation of the tail magnetic field. Some electrons precipitate into the ionosphere (exciting the aurora). However, depending on the pitch angle, other electrons may, by conservation of magnetic moment, undergo reflection at the magnetic mirror points. These electrons travel upwards towards the auroral plasma cavity (e.g., Benson and Calvert, 1979; Calvert, 1981; Ergun et al., 1998; Mutel et al., 2008), where there is not enough plasma to contain

the energy of the incoming electrons (e.g., Treumann and Baumjohann, 2020) (plasma density is sufficiently low that collisions are rare). As a result, the electrons undergo wave-particle interactions, and emit their energy via the Electron-Cyclotron Maser Instability (ECMI) as AKR. This simplified view does not account for particles with a variety of pitch angles which contribute to the instability, for example downgoing electrons.

AKR is anisotropically beamed in a hollow cone at angles near-perpendicular to the source region (Wu and Lee, 1979;

Wu, 1985), although some undergo strong refraction along the inner edges of the auroral cavity which adds complexity to the beaming (Mutel et al., 2008; Menietti et al., 2011). This generates a statistical illumination region based on radio sources in both hemispheres, with generally right-handed circularly polarized emission from the northern magnetic (southern geographic) hemisphere, and left-handed circularly polarized emission from the southern magnetic (northern geographic) hemisphere. The manner in which these two sources interact generates a statistical shadow zone at equatorial magnetic latitudes close to the

planet (inside $\approx 5$ Earth radii on the nightside e.g., Morioka et al., 2011), where AKR is not observed, and an observer at higher latitudes or further distance may see one or other source, or a superposition of both. This anisotropic beaming creates challenges for observing AKR, since a spacecraft orbiting the Earth will transit into and out of the illumination regions, as well as observing changes in AKR emission as a result of solar wind - magnetosphere coupling.

Although AKR has been observed at all local times (LTs) (Zhao et al., 2019; Waters et al., 2021b; Fogg et al., 2022), it is

most reliably viewed on the nightside, between 18 and 6 LT (e.g., Gurnett, 1974; Green et al., 1977; Kasaba et al., 1997; Zhao et al., 2019; Fogg et al., 2022), relating to the statistical position of the source region. The observed power decreases as $\frac{1}{R^2}$ (Gurnett, 1974; Green et al., 1977), resulting in higher powers being observed closer to the source region. Finally, a 24 hour modulation of the AKR signal has been observed, as a result of the diurnal precession of the tilted dipole magnetic field (Lamy et al., 2010; Panchenko et al., 2009; Morioka et al., 2013; Waters et al., 2021b).

Previous AKR observations have highlighted the link between the emissions and geomagnetic activity. The intensity and frequency range of AKR has been shown to correlate with the AE index (Dunckel et al., 1970; Voots et al., 1977; Hashimoto et al., 1998), which is often explained by its strong link to substorm activity (Morioka et al., 2007, 2011, 2014; Waters et al., 2022). Similarly, AKR enhancements are sometimes observed simultaneously to auroral brightenings (e.g., Gurnett, 1974). More broadly, longitudinal extensions of the source region are observed with increasing geomagnetic activity (Zhao et al.,





2019), enabling AKR observations at dayside LTs. Additionally, Fogg et al. (2022) showed higher geomagnetic activity in AU, AL, PC(N) and SYM-H during AKR bursts, and even greater enhancements in activity during the most intense AKR events. Finally, (Kurth et al., 1998) showed that prolonged southward IMF $B_Z$ enhanced AKR emission during the passing of a magnetic cloud event.

The terrestrial ionosphere is permeated with interwoven current systems, which are a barometer for energy transfer through-
out the magnetosphere. As with any electrical circuit, when one current is excited, those that connect with it respond accordingly, ensuring current continuity. In this study, the horizontal currents in the polar and high latitude ionosphere are characterised using the polar and auroral indices. For a detailed description of the role of ionospheric currents in the magnetosphere, the reader is directed to: Milan et al. (2017); Cowley (2000) (and references therein).

In this study, the geomagnetic indices AE, AU, AL, PC(N) and SYM-H (obtained via OMNIWeb (Papitashvili and King,
2020)) are utilised to examine coupling timescales within the magnetosphere. Using these geomagnetic indices is favourable as they offer continuous (preferable in the technique used), minute resolution data.

The PC-index (Troshichev and Andrezen, 1985; Stauning, 2013) monitors activity in the polar ionosphere, and in the northern geographic (containing a southern magnetic pole) hemisphere is derived from a magnetometer station at around 85° latitude in Greenland. Geomagnetic observatories such as these measure deflections in the magnetic field as a result of changes in over-
head current systems, and as such the PC-index can be used to probe the state of polar ionospheric currents. At such latitudes, a horizontal Hall current (pointing sunwards) characterises the speed of magnetic flux transport across the polar cap (e.g., Milan et al., 2017). Although the PC-index is not an exact measure of the strength of these currents, an enhancement in the PC-index demonstrates an excitation of these polar currents, and hence an intensification of the tailward magnetic flux transport. For example, following the onset or enhancement of dayside reconnection, an increase in PC may be observed as flux builds up
in the tail prior to substorm onset. Additionally, the PC index responds to changes in solar wind dynamic pressure (e.g., Fogg et al., 2023a) and magnitude of the interplanetary magnetic field Troshichev et al. (2021), and as such is often regarded as a monitor of solar wind energy input into the magnetosphere.

At auroral latitudes, the commonly used auroral electrojet index (AE, Davis and Sugiura, 1966; World Data Center for Geomagnetism Kyoto et al., 2015) and its upper and lower envelopes (AU and AL respectively) characterise the strength
of activity in the auroral zone. Similarly to the PC-index, they are derived from magnetometer stations at auroral latitudes, which observe deflections in the geomagnetic field as a result of overhead currents, in this case the auroral electrojets. Notably an indicator of substorm activity, AL indicates the strength of westward electrojets, including the substorm electrojet (the ionospheric portion of the substorm current wedge). Similarly, AU indicates activity in eastward electrojets.

Finally, SYM-H is derived from magnetometer stations at equatorial latitudes (Iyemori, 1990), and as such measures the
strength of the ring current. The ring current is a westward flowing equatorial magnetospheric current system formed from a combination of gradient and curvature drift of plasma in the dipole magnetic field. This system is excited by a large deposition of energy in the main phase of a geomagnetic storm. This results in well observed characteristic signatures in the SYM-H index (e.g., Walach and Grocott, 2019), which is often termed the *ring current index*. SYM-H also exhibits step-like increases as a result of rapid magnetospheric compressions known as sudden commencements (e.g., Araki, 1994; Fogg et al., 2023b).



In this paper mutual information (MI) is used to characterise the strength of the relationship between two time series, without any knowledge of the order of their relationship. MI quantifies the correlation, linear or non-linear, between two sequences in a non-parametric (hence model-independent) manner. Although this is an emerging technique in the field of solar system plasma physics, previous authors have demonstrated the use of MI as a powerful tool in understanding non-linear dynamics, and the technique is well-established in the wider community (Shannon, 1949; MacKay, 2003). March et al. (2005) characterised the

MI content between the product of IMF $B_Z$ with solar wind speed, and the auroral electrojet index. They also investigated the effect of various different propagation techniques for the IMF and solar wind data. Wicks et al. (2009) investigated the spatial correlation properties of the solar wind, and used the normalised MI to confirm results without the assumption of linearity. Johnson and Wing (2014) used information theory to show that the northward IMF turnings observed around substorm onset are likely coincidental. Additionally, Snelling et al. (2020) showed a relationship between solar flares and subsequent solar

flares by calculating the MI of time-lagged waiting time distributions of flare events. Finally, Wing et al. (2020) used MI to characterise the coupling timescale between evidence of plasma injections in the Kronian magnetosphere and narrowband emissions.

    In this study, the coupling timescales between the AKR intensity measured by Wind/WAVES and the geomagnetic indices described above (AE, AU, AL, PC(N), SYM-H), as well as IMF $B_Z$ are estimated. The shared information content between

an AKR intensity time series paired with different geomagnetic parameters will be estimated using MI. The geomagnetic indices/IMF are time-lagged with respect to AKR intensity, with lags $-60 <= \tau <= +60$ minutes for indices and $-120 <= \tau <= +30$ minutes for IMF $B_Z$. The lag at which the MI peaks, gives an estimate of the time taken for information from a shared driver to propagate between different regions in the magnetosphere, i.e. the coupling timescale. The method and results of this analysis are described in section 2, followed by concluding remarks.

## 110  2   Estimation of Coupling Timescales

### 2.1   Data and Method

AKR emission is selected from amongst a complex superposition of phenomena in Wind/WAVES/RAD1 (Bougeret et al., 1995) data using the W21 technique (described in detail by Waters et al. (2021b)); a subset of these data are available online (Waters et al., 2021a). Using this W21-selected data, the AKR intensity is integrated over 100-650 kHz as in Waters et al. (2021b); Fogg

et al. (2022) (using the same technique as Lamy et al. (2010, 2008)). Using the same approach as Waters et al. (2021b), AKR intensity is calculated over a full frequency sweep in Wind/WAVES data, roughly 183 seconds (with some fraction of seconds that varies from integration to integration). Since the geomagnetic indices which AKR will be compared with are provided at integer minute resolution, one or other of the datasets must be interpolated to allow point to point comparison. As the resolution of AKR power is uneven and not at integer minutes, and interpolating the indices would be downsampling the data, the AKR

power is interpolated to match the resolution of the geomagnetic indices. This results in AKR intensity measurements and the geomagnetic indices PC(N), AE, AU, AL, SYM-H and IMF $B_Z$ all on the same time scale for comparison. This time series



of AKR intensity has been shown to be correlated to auroral intensity, and so represents a measure of energy input into the ionosphere.

AKR intensity is calculated over ten years of data. AKR observations from 1995 to 2004 inclusive from Wind/WAVES/RAD1
are used, and compared with available geomagnetic indices AE/AU/AL, PC(N), SYM-H and IMF $B_Z$ which are described in detail in the Introduction.

Where the relationship between two parameters is linear, the Pearson correlation coefficient is often used as a measure of the strength and direction of the relationship between two variables (e.g., Vaughan, 2013; Fogg et al., 2020). However, initial investigations, as well as previous studies (e.g., Voots et al., 1977) indicate a more complex relationship between excitations
in auroral indices and AKR power. The mutual information (MI, measured in bits) between two time series is a measure of the relationship between two variables, and is sensitive to both linear and non-linear relationships. Simply put, the MI content between two time series, describes the information learnt about one time series by observing the other (e.g., March et al., 2005), and vice versa.

The standard definition of the MI content, $I(\mathbf{a}, \mathbf{b})$, between two signals $\mathbf{a}$ and $\mathbf{b}$ in terms of entropy, $H$, is (Shannon, 1949;
MacKay, 2003):

$$I(\mathbf{a}, \mathbf{b}) = H(\mathbf{a}) + H(\mathbf{b}) - H(\mathbf{a}, \mathbf{b}) \tag{1}$$

where entropies are defined in terms of the probability distributions of time series a and b. In this study, the mutual information is calculated using the function `mutual_info_regression` (standard options) within the `scikit-learn python` package (Pedregosa et al., 2011), which is based on the method by Kraskov et al. (2004) (following on from work by
Kozachenko and Leonenko (1987)). The Kraskov et al. (2004) estimate for MI is based on $k$-nearest neighbour statistics, and is given by:

$$I(a, b) = \psi(k) - \langle \psi(n_a + 1) + \psi(n_b + 1) \rangle + \psi(N) \tag{2}$$

where $\psi$ is a digamma function, $k$ is the number of nearest neighbours, $n_a$ is the number of measurements in a, $n_b$ is the number of measurements in b, and $N$ is the number of measurements overall. It is important to note at this point that
the variables a and b used in this manuscript are discrete samples of continuous variables derived from empirical observations from both ground based magnetometers and a space based radio instrument. Rather than varying from -1 to 1 as for the Pearson correlation coefficient, $I$ is positive, with higher values indicating more shared information content between two time series.

The Kraskov et al. (2004) method is used here as it has some advantages to the traditional method. In particular, it avoids bias which comes from the binning of of the data into probability distribution functions. Additionally, the Kraskov et al. (2004)
method is 'data efficient', resolving structure to the smallest scales, and 'adaptive', meaning that the resolution improves as the amount of data increases. The reader is directed to Kraskov et al. (2004) for further details.

Random phase surrogates are a widely used method to test the results against the null-hypothesis that there is no time-structure in the time series. In order to assess the significance of calculated MI values, each calculation is repeated using



a random phase surrogate of AKR intensity. The surrogate AKR intensity is generated using an amplitude adjusted fourier

transform (Schreiber and Schmitz, 1996), which is calculated using the `aaft` python package. The resulting surrogate AKR retains the amplitude information, but not the temporal information in the AKR time series: both the surrogate and observed AKR intensities have very similar PDFs (e.g., Tindale et al., 2018). Since the surrogate AKR has no temporal relationship with magnetospheric activity, MI values between the surrogate and geomagnetic indices should be lower than for the real AKR intensity: the surrogate is used to quantify the MI content in correlations that occur 'by chance'.

While using the MI estimate devised by Kraskov et al. (2004) (Equation 2), this investigation exploits a similar overall methodology to that employed by March et al. (2005), who gives a detailed description of MI calculation and estimated the coupling timescales between the AE index and $v_x B_Z$. Also comparing different propagation methods for $v_x$ and $B_Z$ measurements from the Wind spacecraft to Earth, they calculated the MI between the two time series for different values of 'additional time lag', which characterised the response time of the magnetospheric system.

In this study, a similar time lag approach is used to assess the strength of the relationship between time series of AKR and geomagnetic indices.

## 2.2 Results

### 2.2.1 MI coupling analysis

Similarly to March et al. (2005), the MI content between time series of AKR power and geomagnetic indices with different $\tau$

are calculated, and presented as black crosses in Figure 1. Note that the AKR power timestamps are fixed, and geomagnetic indices time series are shifted by $-60 <= \tau <= +60$ minutes, while IMF $B_Z$ is shifted by $-120 <= \tau <= +30$ minutes. For each parameter, a piecewise linear (purple) and quadratic (gold) fit to the MI content between time series (black crosses) is performed, and the peak of each of those fits is indicated with a dashed vertical line in the corresponding colour. The piecewise linear curve is fit to the data as one curve with free parameters including the position of the turnover. For all parameters except

PC(N) and SYM-H the piecewise linear fit to the data has a lower root-mean-squared (RMS) error than the quadratic fit; although the RMS values are of the same order of magnitude for both fits in AU, PC(N) and SYM-H.

To determine whether the calculated MI values are significant, they are compared to the MI content between the surrogate AKR and the indices (see description in section 2.1). The mean of MI values between the surrogate AKR and each index across all lag times is indicated with a blue arrow on each panel of Figure 1. For all indices, this 'threshold' value falls at least an

order of magnitude below the MI represented by the black crosses, confirming that the MI between the AKR and indices is statistically significant.

It is important to note that this analysis was also run on IMF total magnitude and $B_Y$, as well as solar wind flow pressure, density and speed. No clear trend in MI as a function of lag was found, and indeed sometimes the MI was below the random phase surrogate threshold. Therefore only results for parameters with clear MI vs lag trends (i.e. AE, AU, AL, PC(N), SYM-H

and $B_Z$) are presented for brevity. Additionally, any time intervals with missing data (which are only found in PC(N) and $B_Z$) were removed from both the parameter data set and the AKR dataset before running the MI-lagging analysis.



Before discussing the MI results, it is important to note that this time-lagging technique was also applied to the data using the Pearson correlation coefficient, $r$, rather than MI, finding a line of best fit and its related $r$ value. The Pearson correlation coefficient is a measure of the strength and direction of a linear relationship between two variables. $r$ varies from -1 to 1, where the sign indicates whether the relationship is positive or negative. Values of $r$ close to 0 indicate that y may increase or decrease as x increases (Vaughan, 2013). Conversely, values of $r$ close to +1 (-1) suggest that y will increase (decrease) as x increases. For all geomagnetic indices, the value of $r$ across all lag times was below 0.15. This suggests the linear correlation between the data are low - since the values are much closer to 0 than 1. This emphasises the non-linear nature of the relationship between AKR intensity and geomagnetic indices (e.g., Voots et al., 1977), and the need for a non-linear measure of correlation, such as MI.

For the auroral indices, the peak MI content between the indices and AKR power is within +/-10 minutes. Taking only the peak of the better fitting curve, each of the auroral indices has the most shared information with the AKR power with a small positive temporal lag. Peaks are found at +7 minutes for AE (Figure 1(a)), +1 minutes for AU (Figure 1(b)), and +8 minutes for AL (Figure 1(c)). Errors on these values are found at the intersection of a horizontal line threading the peak and the edges of the 80% confidence interval, represented as a shade, and are presented in table 1. Given previous work suggesting that AKR excitations are driven by the same phenomena which excite the auroral zone (which is characterised by AE/U/L), this suggests that the AKR source region (between 1.8 $R_E$ and 3 $R_E$ in altitude (Morioka et al., 2007; Calvert, 1981)) feels these effects before the auroral region. It is interesting to note that the peak MI is much better defined for AE and AL, perhaps as a result of the strong relationship between AKR and substorm activity characterised by AL (unlike the study by (Lamy et al., 2010) which showed the best correlation between AKR and AU). Additionally, the error on the calculated AU peak is larger (as detailed in table 1), and excitations in AKR and AU are more closely temporally aligned.

The MI content between the northern geographic hemisphere polar cap index (PC(N)) and the AKR power is displayed in Figure 1(d). The PC(S) index is omitted to avoid interhemispheric asymmetries as a result of seasonal variations in ionospheric conductance. For both fits, the MI peaks at a negative lag $\geq -5$ minutes. For the slightly better fitting quadratic fit, this suggests that the PC(N) index is excited 11 minutes before any corresponding enhancement in AKR power (the piecewise fit suggests $\tau = -5$ minutes). This suggests that 11 minutes before an excitation in the AKR power, a corresponding enhancement in the transfer of flux across the polar cap will be observed. This may relate to some portion of the building up of flux in the magnetotail preceding substorm onset, which takes on the order of 30-90 minutes (e.g., Li et al., 2013), and its related expansion onset has been shown to drive AKR enhancements. Alternatively, it could relate to the increase in AKR power in the 20 minutes prior to substorm onset shown by Waters et al. (2022).

The ring current index SYM-H is compared with AKR intensity at different time lags, and presented in Figure 1(e). Although the best fit to these data is the quadratic curve (gold), the RMS error is similar for both fits. Compared with the other indices, there is greater variability across different time lags. The calculated peak varies from +3 (piecewise fit) to +9 ($x^2$ fit) minutes, but both have large error values, which are detailed in table 1. This may suggest that a shared driver between AKR and SYM-H may excite AKR emissions before the enhancement of the ring current (or indeed triggering a geomagnetic storm). It is important to note that the random phase surrogate threshold is only one order of magnitude below MI values for SYM-H.





**Table 1.** Fit parameters and errors on $x^2$ and piecewise linear curve fitting to mutual information data.

| | $x^2$ curve $-a(x+b)^2 + c$ errors on a,b,c | $x^2$ RMS | $x^2$ peak error (minutes) | piecewise curve $k_1 x + y_0 - k_1 x_0 \; x < x_0$ $k_2 x + y_0 - k_2 x_0 \; x >= x_0$ errors on $k_1, k_2, x_0, y_0$ | piecewise RMS | piecewise peak error (minutes) |
|---|---|---|---|---|---|---|
| AE | $a = 4.1 \times 10^{-6}, b = 2.0, c = 0.05$ $\pm 9.2 \times 10^{-8}, \pm 0.36, \pm 0.0002$ | $1.18 \times 10^{-6}$ | -2 -18.6,+18.6 | $k_1 = 0.0002, k_2 = -0.0003, x_0 = 6.9, y_0 = 0.05$ $\pm 3.0 \times 10^{-6}, \pm 4.1 \times 10^{-6}, \pm 0.32, \pm 8.6 \times 10^{-5}$ | $2.13 \times 10^{-7}$ | +7 -3.2,+1.82 |
| AU | $a = 1.8 \times 10^{-6}, b = 1.0, c = 0.03$ $\pm 3.5 \times 10^{-8}, \pm 0.3, \pm 5.7 \times 10^{-5}$ | $1.7 \times 10^{-7}$ | -1 -17.1,+17.1 | $k_1 = 0.0001, k_2 = -0.0001, x_0 = 1.5, y_0 = 0.03$ $\pm 2.9 \times 10^{-6}, \pm 3.1 \times 10^{-6}, \pm 0.7, \pm 7.5 \times 10^{-5}$ | $1.6 \times 10^{-7}$ | +1 -4.9,+5.2 |
| AL | $a = 4.5 \times 10^{-6}, b = 0.9, c = 0.04$ $\pm 1.1 \times 10^{-7}, \pm 0.4, \pm 0.0002$ | $1.7 \times 10^{-6}$ | -1 -19.3,+19.4 | $k_1 = 0.0002, k_2 = -0.0004, x_0 = 7.8, y_0 = 0.05$ $\pm 2.8 \times 10^{-6}, \pm 4.0 \times 10^{-6}, \pm 0.3, \pm 8.1 \times 10^{-5}$ | $1.9 \times 10^{-7}$ | +8 -2.91,+1.55 |
| PC(N) | $a = 2.2 \times 10^{-6}, b = 11.3, c = 0.03$ $\pm 4.4 \times 10^{-8}, \pm 0.4, \pm 7.0 \times 10^{-5}$ | $2.7 \times 10^{-7}$ | -11 -17.6,+17.0 | $k_1 = 0.0001, k_2 = -0.0002, x_0 = -5.2, y_0 = 0.03$ $\pm 4.6 \times 10^{-6}, \pm 3.5 \times 10^{-6}, \pm 0.7, \pm 0.0001$ | $2.9 \times 10^{-7}$ | -5 -6.8,+4.0 |
| SYM-H | $a = 6.3 \times 10^{-7}, b = -9.2, c = 0.02$ $\pm 3.1 \times 10^{-8}, \pm 0.9, \pm 4.9 \times 10^{-5}$ | $1.3 \times 10^{-7}$ | +9 -27.1,+27.4 | $k_1 = 4.8 \times 10^{-5}, k_2 = -3.1 \times 10^{-5}, x_0 = 3.0, y_0 = 0.03$ $\pm 2.7 \times 10^{-6}, \pm 3.0 \times 10^{-6}, \pm 1.8, \pm 7.2 \times 10^{-5}$ | $1.4 \times 10^{-7}$ | +3 -10.2,+16.0 |
| $B_Z$ | $a = 2.1 \times 10^{-6}, b = 59.5, c = 0.02$ $\pm 5.7 \times 10^{-8}, \pm 0.7, \pm 0.0001$ | $1.4 \times 10^{-6}$ | -60 -26.3,+27.3 | $k_1 = 0.0001, k_2 = -0.0002, x_0 = -52.6, y_0 = 0.0252$ $\pm 4.1 \times 10^{-6}, \pm 3.1 \times 10^{-6}, \pm 0.6, \pm 0.0001$ | $4.3 \times 10^{-7}$ | -53 -6.1,+4.8 |





Despite the large error values on the peak, and the variability of the MI content across different lags, the MI content is above the threshold value set by the random phase surrogate, so may be interpreted as significant. Since AKR sources are generally centred at high latitudes (e.g., Calvert, 1981; Johnson et al., 2001), and the SYM-H index is measured by near equatorial magnetometers, a strong relationship between the two was not expected. However, there may be more to investigate, for example an assessment of AKR power with respect to storm phases detected in SYM-H (such as those presented by Walach and Grocott (2019)) could allude to any relationship between AKR intensity and ring current activity.

Finally, bow shock IMF $B_Z$ is compared with AKR intensity, at different values of lags to geomagnetic indices, and is presented in Figure 1(f). Lags extending to more negative values are used because it is expected that changes in dayside IMF $B_Z$ may be considerably earlier than changes in AKR, a predominantly nightside phenomena. The MI as a function of applied temporal lag is fit best by the piecewise linear fit, with a peak at -53 minutes, and a relatively tightly fitting confidence interval. This suggests that any change in IMF occurs some 53 minutes before any related AKR excitation or enhancement. This is in keeping with canonical timescales for propagation of dayside changes over to nightside regions, i.e. substorm growth phases (e.g., Forsyth et al., 2015).

It is important to note at this point that for all parameters, the calculated MI values are below 0.05 for all temporal lags. Although the significance of these values was tested using the random phase surrogate threshold, a discussion of potential reasons for these low values will be given here. Firstly, the geomagnetic indices and AKR intensity, although both measures of magnetospheric activity, are sampling different regions in the magnetosphere, particularly in terms of altitude. Additionally, due to the anisotropic beaming of AKR, its visibility and observed intensity also varies dramatically with latitude and local time (e.g., Morioka et al., 2011; Fogg et al., 2022; Waters et al., 2022). The AKR intensity data have not been corrected for this variation (as doing so is a non-trivial open question), which may contribute to low MI values. Finally, the generation of AKR intensity and geomagnetic indices is very different: AKR intensity is observed by a single spacecraft moving through the magnetosphere, whereas the indices (excluding PC(N)) are averaged over multiple stations, so there may be some smoothing effects contributing to low MI values.

### 2.2.2 Application of coupling timescale

The results presented in section 2.2.1 obtain a statistical estimate for the time-lag between AKR intensity and geomagnetic indices. These were estimated over ten years of AKR intensity data, and as such represent a broad overall relationship between the two time series. To investigate this further, examples of substorms from November 2003 (where Wind was on the nightside of the planet and hence in the statistical AKR visibility region (e.g., Wilson III et al., 2021)) were studied and the relationship between AKR power and indices was examined. One such example is presented in Figure 2. Substorm onsets were extracted from the Substorm Onsets and Phases from Indices of the Electrojet (SOPHIE, Forsyth et al., 2015) 75% list, which determines substorm phase from percentiles of rates of change of SuperMAG AL (SML, equivalent to AL).

A frequency-time-intensity spectrogram of W21-selected data is presented in panel 2(a), with a black dashed line indicating substorm onset. AKR intensity begins to increase just before substorm onset (similarly to results by Waters et al. (2022)), and this enhancement is also observed in a time series of AKR integrated intensity presented in Figure 2(b). Additionally, the AKR



band expands to lower frequencies shortly following substorm onset, although there is also some random noise uncorrelated in time and frequency around this low-frequency extension (LFE).

Focussing first on the auroral indices, AU, AL and AE are presented in purple, green and black respectively in Figure 2(c). AL shows a small, but characteristic substorm signature starting with a rapid decrease shortly after the indicated substorm onset,

but simultaneously to the green dotted line. The green dotted line indicates the substorm onset plus the AL lag of maximum MI (+8 minutes). This example shows that although AKR is excited around substorm onset, the AL index begins its excitation around 8 minutes later. AE shows a similar enhancement, dominated by the AL signature.

The polar cap index PC(N) and ring current index SYM-H are presented in pink and black respectively in panel 2(d). The pink dotted line is drawn at substorm onset plus the PC(N) lag of maximum MI (-11 minutes). At the point at which the

pink dotted line is drawn, PC(N) begins to reduce into trough. Recalling that MI assesses the relationship between variables independent of the order and direction of the relationship, this may be an indication of PC(N) decreasing about 11 minutes before an AKR intensity enhancement. The lag of maximum MI from both fits (substorm onset +3 to +9 minutes) for SYM-H are presented as grey shade in panel 2(d). This coupling indicator time period doesn't coincide with any obvious indicators of clear ring current activity, and thus doesn't help to unravel the relationship between AKR and SYM-H.

Finally, IMF $B_Z$ is presented in orange in Figure 2(e). The orange dotted line indicates substorm onset (dashed black line) minus 53 minutes - the calculated lag of maximum MI. This indicator coincides with a sharp southward IMF turning. This may relate to an enhancement in magnetic reconnection, leading to flux building up in the tail prior to substorm onset (and indeed AKR excitation).

## 3 Conclusions

In this study the MI content between AKR intensity and various relevant geomagnetic indices as well as IMF $B_Z$ at different temporal lags has been assessed. Both quadratic and piecewise linear functions were fitted to the MI as a function of lag data. The time lag at which maximum non-linear correlation occurs has been extracted and interpreted as the coupling timescale between AKR intensity and the phenomena represented by each geomagnetic index or $B_Z$. It is important to note at this point that correlation does not necessary imply causality, however the wealth of literature on this subject suggests strong links

between indices and AKR. The physical implication of this analysis and key results are summarised below:

1. The AKR source region feels the effects of a shared driver before the auroral ionosphere

2. This delay is more noticeable for westward electrojets (including the substorm electrojet) characterised by AL than eastward electrojets characterised by AU

3. For AE / AU / AL the lag of maximum MI is +7 / +1 / +8 minutes

4. The PC(N) index is excited 5-11 minutes before any corresponding enhancement is observed in AKR



5. This suggests an enhancement in antisunwards flux transport preceding AKR intensity: this confirms previous suggestions of flux building up in the magnetotail, leading to substorm onset which excites AKR emission

6. The relationship between temporal lag and mutual information between SYM-H and AKR intensity is less clear, but may suggest that the AKR intensity feels the effect of a shared driver before the ring current

7. IMF $B_Z$ changes 53 minutes before any related AKR excitation

Further work to understand the coupling timescales between the AKR source region and the ionosphere could include a parameterisation of this analysis by different frequency regions within the AKR emission. Since emission frequency is inversely proportional to source altitude, this could provide more information on the propagation of driver effects along a magnetic field line. Additionally, further parametrisation by local time could help to explain viewing effects on these coupling timescales.

*Code and data availability.* Wind/WAVES data that has been empirically selected for AKR emissions using the technique by Waters et al. (2021b), and a subset is available online (Waters et al., 2021a). Geomagnetic indices AE, AU, AL, PC(N), and SYM-H as well as $B_Z$ were obtained via OMNIWeb (Papitashvili and King, 2020; Papitashvili, N., 2023). We gratefully acknowledge use of NASA/GSFC's Space Physics Data Facility's OMNIWeb service, and OMNI data. The AU, AL and SYM-H indices used in this paper were provided by the WDC for Geomagnetism, Kyoto (http://wdc.kugi.kyoto-u.ac.jp/wdc/Sec3.html) via OMNIWeb. PC(N) index was provided by World Data Center
for Geomagnetism, Copenhagen via OMNIWeb. We acknowledge the use of python packages scikit-learn (Pedregosa et al., 2011), aaft (https://github.com/lneisenman/aaft), and scipy (https://docs.scipy.org/doc/scipy/index.html). Code to estimate the coupling timescales using mutual information is available from Fogg (2023).

*Author contributions.* ARF, CMJ and SCC designed the method. ARF coded up the method with testing from AB. ARF created all plots with input from CMJ and SCC. JEW processed the data using the W21 technique. ARF prepared the manuscript with contributions from all
co-authors.

*Competing interests.* The authors declare that they have no conflict of interest.

*Acknowledgements.* A.R.F.'s work was supported by Irish Research Council Government of Ireland Postdoctoral Fellowship GOIPD/2022/782 and Science Foundation Ireland Grant 18/FRL/6199. C.M.J.'s work was supported by the Science Foundation Ireland Grant 18/FRL/6199. S.C.C. acknowledges AFOSR grant FA8655-22-7056 and an ISSI Johannes Geiss Fellowship. J.E.W.'s work was supported by the EPSRC
Centre for Doctoral Training in Next Generation Computational Modelling Grant No. EP/L015382/1. The authors acknowledge CNES (Centre National d-Etudes Spatiales), CNRS (Centre National de la Recherche Scientique) / INSU (Institut national des sciences de l'Univers)



programs of planetology and heliophysics, and Observatoire de Paris for support to the Wind/WAVES team and the CDPP (Centre de Données de la Physique des Plasmas) for the provision of the Wind/WAVES RAD1 L2 data.



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





**Figure 1.** MI content between the labelled parameter and AKR integrated intensity as a function of time lag between the two variables (black crosses). Lag time is the time shifted applied to the geomagnetic index data. Parameters shown (a) AE (b) AU (c) AL (d) PC(N) (e) SYM-H (f) IMF $B_Z$. A piecewise linear (purple) and quadratic (gold) fit to the MI data is presented, with the position of the MI peak indicated with a dashed line in the corresponding colour, labelled with the x position. The corresponding shade represents the 80% confidence interval. The mean of the MI content between each index and a random phase surrogate of AKR intensity is indicated with a blue arrow, not in scale with the y-axis where necessary.







**Figure 2.** (a) Frequency-time-intensity spectrogram of W21-selected (Waters et al., 2021b) AKR data observed by the Wind satellite with time series of (b) AKR integrated intensity between 100 and 650 kHz, (c) Auroral Electrojet Index (AE, black) and it's upper (AU, purple) and lower (AL, green) envelopes; green dotted line indicates AL coupling timescale from MI analysis, (d) ring current index (SYM-H, black) and polar cap index (PC(N), pink); pink dotted line indicates PC(N) coupling timescale, grey shade indicates SYM-H coupling timescales from both fits, (e) IMF $B_Z$ in GSM coordinates; orange dotted line indicates the coupling timescale for $B_Z$. In all panels, black vertical dashed line denotes substorm onset from the SOPHIE 75% substorm list.