# Peer review of "Quantification of Magnetosphere - Ionosphere coupling timescales using mutual information: response of terrestrial radio emissions and ionospheric/magnetospheric currents"

_EGUsphere, 2023_

## Author Response (AR1)

In this document, text from the reviewers comments is in *blue italics*, and our response is in plain black text. In the redlined version of the manuscript, our changes are in bold red font.

**RESPONSE TO RC1 ('Comment on egusphere-2023-2055', Rajkumar Hajra)**

*Comments on manuscript (# egusphere-2023-2055): "Quantification of Magnetosphere – Ionosphere coupling timescales using mutual information: response of terrestrial radio emissions and ionospheric/magnetospheric currents" by Alexandra Ruth Fogg et al.*

*The manuscript presents an interesting study on the relationship of the auroral kilometric radiation (AKR) with several auroral indices, symmetric ring current index SYM-H, and interplanetary magnetic field (IMF) Bz component. The technique of mutual information is applied to identify that auroral activities lag the AKR excitation by 1 to 8 minutes, SYM-H index lags by 3 to 9 minutes, polar cap index is excited 11 minutes before any corresponding enhancement in AKR power, and IMF Bz changes occur 53 minutes before any related AKR excitation. The results are interesting, and the manuscript is well written. I can recommend publication of the manuscript after a few minor corrections, as suggested below.*

Thank you for your time considering our manuscript, our point by point responses are included below.

1. *Lines 8 – 9: "Bow shock IMF BZ is excited about an hour before AKR enhancements." – Mention full form of IMF.*

Thank you for this correction, we have amended as you suggest. Please see line 9.

2. *In the abstract, include 1-2 sentence/s stating the physical significance of the results.*

Thank you for this suggestion, we have included the following sentences at the end of the abstract: "This work provides quantitatively determined temporal context to the coupling timelines at the Earth. The results suggest that there is a sequence of excitation following onset of a shared driver: first, the polar ionosphere feels the effects, followed by the AKR source region, and then the auroral ionosphere." Please see line 9.

3. *Line 44: "local times (LTs)" should be "LTs", as it is already defined in Line 22.*

Thank you for spotting this error, we have corrected it. Please see line 48.

4. *Lines 46 – 47: "The observed power decreases as 1/R2 (Gurnett, 1974; Green et al., 1977), resulting in higher powers being observed closer to the source region." – Add a sentence that R represents the distance from the source region.*

We have added a note to this effect, thank you for this comment. Please see line 50.

5. *Lines 57 – 58: "Finally, (Kurth et al., 1998) showed that prolonged southward IMF BZ enhanced AKR emission during the passing of a magnetic cloud event." – Mention full form of IMF, define "magnetic cloud", and clearly state the significance of this finding. This will make the manuscript easier to follow for the wider readership.*

We have defined the full form of the interplanetary magnetic field, please see line 61.

We have added a sentence to highlight the findings of the Kurth 1998 study: "Despite limitations with single spacecraft viewing, this study highlighted links between IMF BZ driving and the excitation of AKR emissions." Please see line 62.

6. *Lines 64 – 65: "In this study, the geomagnetic indices AE, AU, AL, PC(N) and SYM-H (obtained via OMNIWeb (Papitashvili and King, 2020)) are utilised to examine coupling timescales within the magnetosphere." – In this statement, what is the importance of citing "Papitashvili and King, 2020" for OMNIWeb? OMNIWeb is a publicly available NASA site proving solar and magnetospheric data. I suggest replacing the reference by the URL address.*

Thank you for this comment. We have left the reference to OMNI data in the original citation format, as suggested by the NPG website: "A data citation in a publication resembles a bibliographic citation and needs to be included in the publication's reference list" (see https://www.nonlinear-processes-in-geophysics.net/policies/data_policy.html). We feel this format most reflects the NPG guideline quoted, and the link to the data is included in the reference list.

7. *Lines 65 – 66: "Using these geomagnetic indices is favourable as they offer continuous (preferable in the technique used), minute resolution data." – Very strange argument! Surely you are not using "these geomagnetic indices" as they are continuous and minute resolution, but for their relationship with AKR. I suggest re-phrasing the statement.*

This is an important point. We have removed this statement, and added the following argument: "These geomagnetic indices are chosen for the study as the characterise auroral, polar and equatorial activity respectively, and hence map to different regions of the magnetosphere. Comparing and contrasting their relationships with AKR will give a global view of AKR's role in the magnetosphere." Please see line 70.

8. *Lines 73 – 75: "For example, following the onset or enhancement of dayside reconnection, an increase in PC may be observed as flux builds up in the tail prior to substorm onset." – For "dayside reconnection", please cite Dungey 1961 (https://doi.org/10.1103/PhysRevLett.6.47).*

We have included the citation as you suggested, please see line 81.

9. *Lines 105 – 107: "The geomagnetic indices/IMF are time-lagged with respect to AKR intensity, with lags −60 <= τ <= +60 minutes for indices and −120 <= τ <= +30 minutes for IMF BZ." – Include 1-2 sentences to say why these two different sets of time lags are considered for indices and IMF BZ?*

This is an important point. We have added a few sentences to explain this choice: "The lags for geomagnetic indices and IMF BZ are different due to expected differences in their relationship with AKR. Both the indices, and AKR are measures of geomagnetic activity, and hence will be more closely temporally aligned. IMF BZ however is a driver of the magnetosphere, so there may be a greater time lag between BZ changes and AKR excitation. Finally, in the initial stages of the work, the peak in the MI vs lag data was found to be at lower (more negative) lags for IMF BZ." Please see line 114.

10. *Lines 112 – 114: "AKR emission is selected from amongst a complex superposition of phenomena in Wind/WAVES/RAD1 (Bougeret et al., 1995) data using the W21 technique (described in detail by Waters et al. (2021b)); a subset of these data are available online (Waters et al., 2021a)." – I would suggest proving an URL for the Wind/WAVES data.*

We have left this in the original format in keeping with NPG guidelines, please see our detailed response to your point 6.

> *11. Lines 114 – 117: "Using this W21-selected data, the AKR intensity is integrated over 100-650 kHz as in Waters et al. (2021b); Fogg et al. (2022) (using the same technique as Lamy et al. (2010, 2008)). Using the same approach as Waters et al. (2021b), AKR intensity is calculated over a full frequency sweep in Wind/WAVES data, roughly 183 seconds (with some fraction of seconds that varies from integration to integration)." – I would suggest avoiding "as in"/ "Using the same approach as", and briefly describing the methods for the reader with proper credit to the earlier studies.*

Thank you for this comment. We have made this section more clear – we cite Waters et al (2021b) and Fogg et al (2022) along with our choice of frequency band to integrate the power over. Additionally, we have made it more clear that we are using the integrated power calculation of Lamy et al (2010, 2008). It is fairly standard in our field to cite the appendix of Lamy et al (2008) for the full description of integrated intensity calculation, but we have made this more clear in the new text: "Using this W21-selected data, the AKR intensity is integrated over 100-650 kHz (the same frequency band as Waters et al. (2021b); Fogg et al (2022)), utilising the technique described in detail by Lamy et al. (2010, 2008). AKR intensity is calculated over a full frequency sweep in Wind/WAVES data, roughly 183 seconds (with some fraction of seconds that varies from integration to integration), normalising to 1 AU. For a detailed description of intensity calculation, the reader is directed to Lamy et al. (2010, 2008)." This change is on line 125.

> *12. Lines 121 – 123: "This time series of AKR intensity has been shown to be correlated to auroral intensity, and so represents a measure of energy input into the ionosphere." – This is again a strange argument! Please re-phrase the statement.*

Thank you for this comment. We have removed this statement, please see line 132.

> *13. Lines 124 – 125: "AKR observations from 1995 to 2004 inclusive from Wind/WAVES/RAD1 are used," – Why particularly these 10 years are used? Due to availability issue or other reason? This should be clearly indicated.*

This is an important point. We have added text to explain this choice: "The years 1995 to 2004 are selected as this ranges from just after the launch of Wind (November 1994), to the year when Wind leaves the near-Earth environment and begins its journey to the L1 point. In the interim 10 years, Wind samples a broad parameter space of locations in the Near-Earth environment, providing a range of AKR observations (it is important to note that AKR is strongly anisotropically beamed)." Please see line 137.

> *14. Line 149: change "binning of of the" to "binning of the".*

Thank you for spotting this error, we have removed it. Please see line 163.

> *15. Lines 156 – 157: "both the surrogate and observed AKR intensities have very similar PDFs" – what is "PDFs"?*

We have spelled out this acronym in the text – probability distribution functions – please see line 171.

> *16. Lines 161 – 162: "who gives a detailed description of MI calculation and estimated the coupling timescales between the AE index and vxBZ" – What is vx? State this for the reader.*

We have defined this in the text: "the product of solar wind velocity x-component and IMF BZ". Please see line 176.

> 17. Lines 228 – 229: "Finally, bow shock IMF Bz is compared with AKR intensity, at different values of lags to geomagnetic indices, and is presented in Figure 1(f)." – Are the lags "to geomagnetic indices" or AKR?

We agree this text was unclear. We have amended the text to: "Finally, bow shock IMF BZ is compared with AKR intensity, at lags ranging from -120 minutes to +30 minutes". Please see line 243.

> 18. Lines 241 – 242: "Finally, the generation of AKR intensity and geomagnetic indices is very different:" – "generation" could be replaced by "observation/measurement technique" or something like that.

We have edited the text as you suggested. Please see line 256.

> 19. In the Conclusion section, after each numbered sentence, add a full stop.

We have edited the conclusion numbered points to include a full stop. Please see line 296.

> 20. Lines 295 – 297: "Wind/WAVES data that has been empirically selected for AKR emissions using the technique by Waters et al. (2021b), and a subset is available online (Waters et al., 2021a). Geomagnetic indices AE, AU, AL, PC(N), and SYM-H as well as BZ were obtained via OMNIWeb (Papitashvili and King, 2020; Papitashvili, N., 2023)." – Suitable URL addresses should be mentioned for Wind/WAVES and OMNIWeb instead of references.

We have left this in the original format in keeping with NPG guidelines, please see our detailed response to your point 6.

> 21. Figure 2, panel c: add more legend markings.

Thank you for this comment. We have added more tick marks on the y axis as you suggested.

**RESPONSE TO RC2: 'Comment on egusphere-2023-2055', Anonymous Referee #2, 14 Nov 2023**

*This is an interesting paper about the use of Mutual Information (MI) to determine the effectiveness of nonlinear processes influencing AKR intensity, where the latter is understood to be directly related to magnetic substorm and storm processes in planetary magnetospheres characterized by auroral phenomena. MI in this study is a tool that quantifies the nonlinear correlation of auroral indices to AKR intensity. In this study AE, AU, AL, PC(N), SYM-H), as well as IMF BZ are used as "variables" that relate in a nonlinear way to AKR intensity. MI is related to the entropy of a state variable in a thermodynamic system as the authors explain.*

*I find the study quite interesting from a statistical analysis perspective, which provides useful knowledge to better understand and quantify the role of auroral indices in explaining and predicting the complex and multilayered, time-dependent phenomena of magnetic storms and substorms in planetary magnetospheres. I learned much about how the different auroral indices are defined and what information each might provide as it relates to a specific space physics variable, such as AKR intensity.*

*Overall, I find the manuscript very well written. The figures are well-made with good explanations. The references are comprehensive. The main points of the work are clear and follow from the analysis.*

*I have little expertise regarding MI, but I deem the results interesting and worth publishing, because of the contribution MI can make to the better understanding of global physical processes that combine to generate a magnetic storm.*

Thank you for your kind comments and for taking the time to review our manuscript. Our point-by-point responses follow.

*I have one comment regarding the generation of AKR, near lines 28 to 33:*

*"These electrons travel upwards towards the auroral plasma cavity (e.g., Benson and Calvert, 1979; Calvert, 1981; Ergun et al., 1998; Mutel et al., 2008), where there is not enough plasma to contain the energy of the incoming electrons (e.g., Treumann and Baumjohann, 2020) (plasma density is sufficiently low that collisions are rare)." As a result, the electrons undergo wave-particle interactions, and emit their energy via the Electron-Cyclotron Maser Instability (ECMI) as AKR. This simplified view does not account for particles with a variety of pitch angles which contribute to the instability, for example downgoing electrons."*

*The above statement is somewhat cumbersome and unclear to me. In a few words, the electron plasma distribution inside a density cavity (in the case of Earth) has velocity-space gradients that can make it unstable to the growth of waves.*

Thank you for this comment. We have adapted the text description to include your explanation. Please note that we have kept some of the text, for example mention of the Cyclotron Maser

Instability as these are important points relating to the physics of the system, and hence linking to other papers in the literature. New explanation reads: "Along with particles of other pitch angles, the electrons undergo wave-particle interactions and emit their energy via the Electron-Cyclotron Maser Instability (ECMI) as AKR. Simply put, the electron distribution inside the auroral plasma cavity has a variety of velocity-space gradients that can lead to the generation of waves." Please see line 34.

---

## Author Response (AR2)

We thank the editor for their time reviewing our manuscript.

In this document, text from the editors comments is in *blue italics*, and our response is in plain black text. In the redlined version of the manuscript, our changes are in bold red font.

*The mechanism of the cyclotron maser by Wu and Lee should be made more prominent in the introduction section of the paper. Now it is barely mentioned on line 48 and not in relation to the theory. This should be stated at the very beginning of the Introduction section.*

Thank you for this comment. We agree that the cyclotron maser instability is a key theory in this field and must be emphasised. We have included some text in the first paragraph to note the key points of the theory: "*AKR is generated by the Electron-Cyclotron Maser Instability (ECMI), a seminal theory developed by Wu and Lee (1979). While describing the generation mechanism for AKR emission, this theory also stipulates that AKR emission frequency is inversely proportional to altitude, and is beamed from the source region at near-right angles.*" Please see line 15.

---

## Author Response (AR3)

We thank the editor for their time reviewing our manuscript.

In this document, text from the editors comments is in *blue italics*, and our response is in plain black text. In the redlined version of the manuscript, our changes are in bold red font.

*The paper is acceptable for publication.*

Thank you for considering our manuscript